# Rice Husk Cellulose-Based Adsorbent to Extract Rare Metals: Preparing and Properties

**DOI:** 10.3390/ma16186277

**Published:** 2023-09-19

**Authors:** Askhat Kablanbekov, Svetlana Yefremova, Feruza Berdikulova, Serik Satbaev, Sergey Yermishin, Nurgali Shalabaev, Baimakhan Satbaev, Alma Terlikbayeva, Abdurassul Zharmenov

**Affiliations:** 1National Center on Complex Processing of Mineral Raw Materials of the Republic of Kazakhstan RSE, Almaty 050036, Kazakhstan; kablanbekov_as@mail.ru (A.K.); pheruza_b@mail.ru (F.B.); esv-ret@mail.ru (S.Y.); alma_terlikbaeva@mail.ru (A.T.); jarmen56@mail.ru (A.Z.); 2School of Materials Science and Green Technologies, Kazakh-British Technical University, Almaty 050000, Kazakhstan; 3RSE Astana Branch National Center on Complex Processing of Mineral Raw Materials of the Republic of Kazakhstan, Astana 010000, Kazakhstan; ssatbayev@gmail.com (S.S.); nurgali_s@bk.ru (N.S.); fnc-astana@mail.ru (B.S.); 4The Department of Nanotechnology and Metallurgy, Mechanical Engineering Faculty, Karaganda Technical University, Karaganda 100000, Kazakhstan

**Keywords:** rice husk, cellulose, adsorbent, textural properties, rare metals, rhenium

## Abstract

Adsorption is one of the common stages in the hydrometallurgy of rare metals. Its efficiency is largely determined by the quality of the sorbent used. The purpose of this work was to create an activated sorbent based on rice husk cellulose for the extraction of rhenium from aqueous solutions. Two types of cellulose were obtained by treating rice husk with a 1.5% NaOH solution (alkaline cellulose) and a nitric acid solution in ethyl alcohol (Kürschner and Hoffer cellulose). They were tested by IR, SEM, TA, TPD-MS, and XRD methods. It was found that Kürschner and Hoffer cellulose does not contain lignin and retains structural order to a greater extent. By means of this cellulose carbonization at 600 °C and activation by physical, chemical, and combined methods, a series of sorbents were prepared and studied by different methods. It was determined that the sorbent KHC4-600VA obtained by combined activation of cellulose carbonizate by water vapor at 850 °C, followed by an alkaline treatment, has the best textural characteristics: S—~1200 m^2^·g^−1^, V—1.22 cm^3^·g^−1^, and R—2.05 nm. KHC4-600VA provides 90% recovery of Re (VII) ions from aqueous solutions. According to the Freundlich model, sorption proceeds favorably on the sorbent’s heterogeneous surface.

## 1. Introduction

Rare and rare earth metals are strategically important minerals for all developed countries in the global community [1]. In the context of scientific and technological progress, for example, the development of Industry 4.0 and “green” energy, the demand for these strategic metals is increasing throughout the world. Kazakhstani ores containing rare metals have a complex composition and often combine several rare metals at once (tungsten-molybdenum, niobium-tantalum-titanium). There are no root deposits, but rare metals often accompany copper, iron, and chrome ores, as well as bauxite. Despite the fact that the raw materials are rich, the multicomponent content of rare metals and rare earth elements is low. Therefore, the extraction of rare metals from raw ore materials is usually a multistage and complex technological process [2].

Rhenium is one of the world’s most sought-after rare metals because of its unique physical and chemical properties. It is a ductile, refractory, corrosion-resistant, and oxidation-resistant metal. Rhenium is primarily used in the aviation industry (80% of the world’s production). Adding as little as 5% of rhenium to the heat-resistant alloys increases the strength of gas turbine blades by 15% and extends their useful life by 25%. Platinum-rhenium catalysts increase the speed of high-octane gasoline production by 7–10 times. Tungsten-rhenium thermocouples can measure temperatures up to 2200 °C [3].

The main method of rhenium extraction from domestic raw materials is conversion into man-made products during the processing of copper and molybdenum ores and concentrates [4,5,6,7]. Studies on rhenium extraction through the processing of scrap heat-resistant alloys [8,9,10,11] and spent catalysts [12] are known. Transition and concentration of rhenium into man-made products during high-temperature calcination and/or smelting of copper or molybdenum concentrates proceeds through the stage of formation of volatile rhenium oxide Re_2_O_7_. Depending on the technological mode, the form of rhenium in the anthropogenic products may be different. The metal may be present as perrenate, sulfide, or lower oxide forms resulting from interaction with gas phase constituents or sublimates. Therefore, rhenium recovery methods include oxidative processes for the conversion to water-soluble forms and aqueous leaching followed by sorption and desorption from rhenium-containing solutions.

Adsorption is considered to be one of the rather cheap and simple techniques to apply not only in the purification processes of various aqueous systems but also in hydrometallurgy [13,14,15,16,17,18,19,20,21,22,23,24,25], including rare earth element (REE) technology [26,27,28]. In recent years, the creation of sorbents based on waste materials, in particular those of plant nature, has represented a broad independent research direction [29,30,31,32,33,34,35,36,37,38,39,40,41,42,43]. However, as noted by Anastopoulos et al. in their paper [27], there are practically no review materials on REEs adsorption by biosorbents. This is not coincidental, since there are few research works in this area as well. Awwad et al. [44] studied lanthanum and erbium sorption by active carbon from rice husk. Zafar et al. [45] and Zhu et al. [46] investigated, respectively, the sorption of Ce(IV) by rice husk and La(III) and Ce(III) by granular hybrid hydrogel prepared by the grafting reaction of acrylic acid onto hydroxypropyl cellulose with attapulgite as the inorganic component. Unfortunately, when analyzing the literature sources, no works on rhenium sorption extraction were found, except for our early studies using activated rice husk carbonizate [47].

In the Concept of Industrial-Innovative Development of the Republic of Kazakhstan for 2021–2025, the Government of our country noted that the future of the global economy is to develop the production of rare and rare earth metals [48]. A total of 8% of the world’s rhenium reserves are accumulated in Kazakhstan’s copper and sulfide deposits [49]. The main raw materials are intermediate products of copper production, mainly solutions of the wet gas recovery system (wash sulfuric acid). Rhenium extraction from wash sulfuric acid is implemented on an industrial scale. However, a significant amount of this metal is also distributed in other anthropogenic products, such as sludge, dust, and slag from copper production. These industrial products are currently not involved in industrial processing. The development of an effective sorbent for rhenium extraction from industrial product processing solutions may allow for the expansion of production sources of rhenium and increase its recovery for commercial products.

In this regard, the present work is focused on obtaining an activated sorbent based on rice husk cellulose. As it was shown in paper [50], in some cases it is advisable to isolate rice husk components in order to further utilize them for various purposes. In this study, cellulose from rice husk was isolated by the action of a 1.5% NaOH solution in three steps and by the Kürschner and Hoffer reagent for the purposes of comparison [51]. The obtained material was subjected to carbonization in the atmosphere by exhaust vapor gases. The carbonizate was activated by physical, chemical, and thermal methods. The composition and structure of cellulose isolated from rice husk by both methods were studied by chemical analysis, scanning electron microscopy (SEM), and infrared spectroscopy (IR), respectively. The thermodegradation of cellulose during carbonization was studied by thermal analysis and temperature-programmed desorption mass spectrometry. The X-ray phase composition and X-ray characteristics of the identified phases of cellulose and cellulose-based materials were determined by X-ray phase analysis (XRD). The morphology of cellulose-based material particles was studied by SEM, transmission electron microscopy (TEM) using microdiffraction, and optical microscopy. The low-temperature nitrogen adsorption-desorption (BET) method was used to determine the textural characteristics (pore radius R, total pore volume V, and specific surface area S) of cellulose-based materials. The ability of new materials to sorb rare metals from aqueous solutions was studied based on the example of Re (VII) ion sorption by the sample with the best textural characteristics. A quantitative description of the sorption process was performed with the help of Langmuir and Freundlich isotherm models.

The work shows that the most favorable way of extracting cellulose from rice husk is by treating the feedstock with Kürschner and Hoffer reagent. Silicon dioxide passing into the cellulose composition is almost completely removed in the course of carbonizate chemical activation with a sodium hydroxide solution. It is advisable to treat the carbonizate with alkali after vapor-gas activation with water vapor. This allows for the development of the specific surface area of the obtained sorbent up to ~1200 m^2^·g^−1^ and the increase of the total pore volume up to 1.22 cm^3^·g^−1^ with a pore radius of 2.05 nm. The new sorbent provides 90% recovery of Re (VII) ions from aqueous solutions. The process is most accurately described by the Freundlich model (R^2^ = 0.9745), whose constants indicate that the sorption of Re (VII) ions proceeds favorably on a heterogeneous sorbent surface.

## 2. Materials and Methods

### 2.1. Cellulose Preparation Process

The rice husk (RH) used in this investigation was obtained from an agricultural enterprise in the Kyzylorda region. The main constituents were (wt.%, dry basis): polysaccharides (48–52), lignin (26), and silicon dioxide (14) [52].

The cellulose from the rice husk composition was isolated by different methods:By the action of a 1.5% NaOH solution in three stages according to Beckman’s method [53]. At the first stage, rice husk was treated with a 1.5% sodium hydroxide solution for 48 h at room temperature with a solid-to-liquid (S:L) ratio of 1:20. Then the solution was poured off, and the solid residue was filled with a new portion of the alkaline solution at the indicated solid-to-liquid ratio and refluxed for 6 h. After separation of the liquid, the solid residue was again filled with fresh NaOH solution (S:L ratio of 1:10) and heated to a temperature of 130 °C, at which it was kept for 1 h. Then the obtained residue was washed with hot distilled water to neutralize the reaction of the wash water and filtered on a Burchner funnel. The residue was dried at 105 °C.By the action of the Kürschner and Hoffer reagent [51], consisting of 1 volume of concentrated nitric acid and 4 volumes of ethyl alcohol. One gram of rice husk was treated for 1 h with 25 cm^3^ of Kürschner and Hoffer reagent by refluxing in a water bath. Such treatment was repeated 4 times, after which the residue was filtered on a glass filter and washed with a fresh portion of Kürschner and Hoffer reagent and hot distilled water to neutralize the reaction of the wash water. The residue was dried at 105 °C.

The cellulose composition was determined as described in papers [51,52].

### 2.2. Cellulose-Based Sorbent Preparation 

The preparation of the sorbent was carried out in several steps. In the first step, carbonized material was obtained by cellulose pyrolysis for 30 min at 600 °C. Then the obtained carbonized material was subjected to activation by physical, chemical, and thermal methods, as shown elsewhere [54].

In the case of physical activation, water vapor was used as the activating agent. Activation was carried out for 30 min at 850 °C. For chemical activation, a 70 g·dm^−3^ sodium hydroxide solution was used. The carbonized material was mixed with the alkaline solution at an S:L ratio of 1:10 and boiled for 90 min. The solid residue was then washed with distilled water until the wash water was neutral.

Combined activation processes were also tested. The material after vapor-gas activation with water vapor was treated with the alkaline solution, and conversely, the carbonizate initially subjected to chemical activation was subjected to vapor-gas activation under the above conditions. The material after chemical activation was calcined at 1000 and 1650 °C under an argon atmosphere for 1 h to examine carbon nanotube and graphene-like structure formation because sorbents produced using their derivatives were found to be active in the REE removal processes [27].

### 2.3. Cellulose and Cellulose-Based Materials Characterization

IR spectra were recorded on a Specord M80 spectrophotometer (Carl Zeiss, Jena, Germany) in the form of press tablets with KBr in the range of 4000–400 cm^−1^. Scanning electron microscopy (SEM) and X-ray spectral micro-analysis were performed using a Superprobe 733 micro-analyzer (JEOL, Tokyo, Japan) with an energy-dispersive spectrometer INCA ENERGY (OXFORD INSTRUMENTS, London, England) to determine surface elements. Tabletop Microscope TM4000Plus (Hitachi, Tokyo, Japan) and Optical Microscope Leica DM2500 P (Leica-Microsystems, Wetzlar, Germany) were used as well. Transmission electron microscopy (TEM) was realized using a JEM 100cx transmission electron microscope (JEOL, Tokyo, Japan). Thermal analysis (TA) was performed using a Hungarian Paulik F.-Paulik J.-Erdey L. system derivatograph Q-1500D (MOM, Budapest, Hungary). The smooth heating of the sample was performed to a temperature of 1000 °C with a temperature increase rate of 12 °C min^−1^ in the atmosphere of the exhaust gas. Temperature-Programmed Desorption Mass Spectrometry (TPD-MS) was carried out using a MKh-7304A monopole mass spectrometer (Electron, Sumy, Ukraine) with electron impact ionization, adapted for thermodesorption measurements as described in papers [50,55]. X-ray diffraction patterns were obtained using a DRON-2 computerized diffractometer with modernized collimation with a Cu Kα radiation source, as described in papers [54,56].

The content of volatile substances was determined as the mass loss of the sample upon heating in a muffle furnace at 900 °C for 7 min in a crucible with a lid and without access to air, after deduction of moisture.

Specific surface area (S), pore volume (V), and average pore radius (R) were determined using a NOVA 2200 Surface Area and Pore Size Analyzer (Quantohrome, Boyton Beach, FL, USA).

### 2.4. Re (VII) Sorption Experiments

Batch sorption experiments were performed at room temperature using synthetic aqueous solutions with different concentrations of Re (VII). The solutions were prepared using NH_4_ReO_4_. An adsorbent in the amount of 0.2 g was mixed with 20 cm^−3^ of the prepared solution in an Erlenmeyer flask. The experiments were carried out under constant stirring conditions (150 rpm) at time intervals ranging from 30 to 200 min. The solutions were filtered after the experiments, and residual concentrations of Re (VII) metal ions in the filtrates were analyzed using atomic absorption spectrometry (Agilent AA240FS, Agilent Technologies, Santa Clara, CA, USA).

## 3. Results

### 3.1. Delignification of Rice Husk

The delignification of rice husk was carried out according to Beckman’s method using a 1.5% sodium hydroxide solution. The yield of the residue, which was named alkali cellulose (AC), was 43.4% (Table 1). The composition of alkali cellulose was found to contain more than 84% polysaccharides, 15.5% lignin, and a small amount (0.1%) of silica. The SEM image (Figure 1a,b) shows that removal of silicon dioxide from the outer epidermis of rice husk as a result of alkaline treatment produces surface loosening, and the inner epidermis adopts a characteristic fibrillar structure.

For complete removal of lignin, rice husk was treated with Kürschner and Hoffer reagent in two-, three-, and four-fold repetitions. The residue yield after two-fold treatment was 53.1% (sample KHC2, Table 1), after three-fold treatment was 50.0% (sample KHC3, Table 1), and after four-fold treatment was 49.6% (sample KHC4, Table 1). The determination of lignin in these samples showed that a small amount (1%) still remained after double treatment. Already after three-fold treatment with Kürschner and Hoffer reagent, lignin is completely eliminated. At the same time, the silica content remains practically unchanged (Table 1): 26.4, 27.4, and 25.2%, respectively. Due to its presence, the outer surface of KHC4, unlike AC, retains the corn cob topology (Figure 1c) typical for the original rice husk [52]. The cellulose fibers on the inner surface are covered with a transparent film.

### 3.2. Study of Rice Husk Cellulose by Infrared Spectroscopy

In the IR spectra of rice husk delignification products using Beckman’s method (AC) and the Kürschner and Hoffer reagent (KHC4), a set of bands typical for the vibrations of polysaccharide functional groups [57,58] is observed (Figure 2). The bands in the range of 3500–2900 cm^−1^ are present due to valence vibrations of the groups CH_2_–OH. In the AC sample (Figure 2a), the bands at 1460 cm^−1^ and 1384 cm^−1^ are caused by strain vibrations of CH_2_- and CH groups, respectively, and at 1175 cm^−1^ by antisymmetric valence vibrations of the C–O–C bridge [57]. The characteristic bands of lignin are preserved (two absorption bands of the benzene ring in the range of 1600–1500 cm^−1^) [50,59,60]. The absorption bands in the range of 1650–1400 cm^−1^ were also identified as asymmetric vibrations of C=C aromatic compounds in papers [40,41]. The absorption bands of silicon dioxide in the IR spectrum of the studied material are poorly visible due to its low content (Table 1), although the complex maximum in the range of 1050 cm^−1^ suggests that the C–O–Si bond is still present [61].

No lignin bands are observed in the spectrum of cellulose isolated from rice husk by four-fold treatment with Kürschner and Hoffer reagent (Figure 2b). The absorption bands of silica (1116, 798, and 471 cm^−1^) are clearly labeled. The main maximum of silica (1116 cm^−1^) overlaps the characteristic bands of cellulose, cascading on its left shoulder. The main difference between the investigated sample and alkaline cellulose AC is manifested in the absence of the so-called amorphous band [62] in the range of 900 cm^−1^ and the presence of the maximum (although very weak) of crystalline cellulose at 1320 cm^−1^ [58]. Band splitting in the 2900 cm^−1^ range into two components with maxima at wavenumbers of 2900 cm^−1^ and 2944 cm^−1^ may be caused by a change in the spatial arrangement of the CH_2_–OH groups [63]. The valence vibration band of hydroxide ions is shifted by 50 cm^−1^ to the low-frequency range of the spectrum, which may be due to the increased content of sorbed water. The band at 3359 cm^−1^ characterizes the asymmetric [40] and symmetric valence vibrations of the OH-group of adsorbed water, whose deformation vibrations are characterized by a band at 1637 cm^−1^. The presence of adsorbed water in Kürschner and Hoffer cellulose is also confirmed by the appearance of absorption bands in the low-frequency range at 665 cm^−1^ and 559 cm^−1^, corresponding to the torsional vibrations of water molecules [64].

### 3.3. Study of Cellulose Thermal Degradation in the Process of Carbonization

The thermal degradation of rice husk cellulose was studied by thermal analysis and temperature-programmed desorption mass spectrometry based on the lignin-free sample KHC4 isolated by Kürschner and Hoffer reagent. DTA, DTG, TG (Figure 3), and P-T curves (Figure 4a) indicate the one-step character of its decomposition in the temperature range of 150–350 °C. The maximum rate of the sample weight loss is observed at 260–270 °C. During thermal heating of KHC4 in the low-temperature range, a number of thermal effects are observed on the DTA curve. The endo effect at 100 °C, caused by the release of adsorbed moisture, is successively replaced by two blurred endothermic effects at 250 and 270 °C. This indicates the occurrence of chemical degradation reactions with heat absorption. The weight loss of the initial sample reaches 45%. Further heating of KHC4 is characterized by a series of exo- and endothermic reactions associated with condensation-destructive and oxidative processes in the solid residue. Complete burnout of organic matter occurs at 625 °C. The total weight loss in this case is 69%. The maximum on the P-T curve at 260 °C (Figure 4a) is formed by desorption of pyrolysis products of polysaccharide residues (*m*/*z =* 60, Figure 5a,b) and derivatives of oxygen-containing heterocyclic compounds, such as pyran and furan (*m*/*z* = 128, 126, 98, 96, 84, and 68, Figure 5a,b). The presence of low-molecular-weight ions in the mass spectrum (Figure 5a) is due to the formation of unsaturated aliphatic compounds during pyrolysis.

Pyrolysis of Kürschner and Hoffer cellulose at 600 °C (KHC4-600) leads to a shift of the peak on the P-T curve to the high-temperature range (Figure 4b) due to the formation of hydrocarbons with a high decomposition temperature (>750 °C). However, the pressure of volatile compounds in the decomposition of carbonized cellulose (KHC4-600) is significantly lower due to their low content (5%). Chemical activation with alkali of the carbonized sample (KHC4-600A) leads to an increase in the total pressure of volatile products during pyrolysis of the activated material (Figure 6a). Furthermore, it promotes additional pyrolysis steps in both the low-temperature (up to 400 °C) and high-temperature (500–750 °C) ranges, which were not observed in the case of KHC4-600 sample pyrolysis. Activation by water vapor leads to the appearance of new pyrolysis stages (KHC4-600V, KHC4-600VA, and KHC4-600AV, Figure 6b–d). The lowest pressure values of gaseous pyrolysis products are observed for samples KHC4-600A-1000 and KHC4-600A-1650 subjected to higher heat treatment (Figure 6e,f).

The mass spectra of pyrolysis products of cellulose carbonizates subjected to different activation methods show a meager set of lines and no high-mass ions (Appendix A). This indicates the occurrence of deep destructive processes during the treatment methods undertaken. At temperatures around 100 °C, desorption of physically sorbed molecules, mainly water with *m*/*z* 18, occurs (Appendix A). The high-temperature maxima observed on the P-T curves at temperatures above 600 °C (Figure 6) are formed due to the release of CO (*m*/*z* = 28) and H_2_ (*m*/*z* = 3) (Appendix A). CO_2_ desorption (*m*/*z* = 44, Appendix A) contributes to the formation of maxima observed at temperatures of 200–600 °C on the P-T curves (Figure 6). After high-temperature treatment of the samples (KHC4-600A-1000 and KHC4-600A-1650), the mass spectra line intensities drop significantly (Appendix A). Correspondingly, as noted above, the pressure value of volatile products decreases significantly (Figure 6e,f). This is a natural consequence of the carbonization process, which removes decomposition products from the samples.

### 3.4. X-ray Phase Composition of Cellulose and Cellulose-Based Materials

The X-ray diffraction pattern of KHC4 (Figure 7) shows two intense reflections with d_002_~4.0 Å and d_hkl_~5.4 Å at angles of 2Θ = 22° (002) and 16° (10ī and 101 not resolved), respectively. These reflections indicate the presence of crystalline cellulose (C_c_) in amounts up to 35%. The transverse size of the crystallites (L_c_) is 55 Å. The presence of a diffuse halo (A_f_) is due to the amorphous cellulose component and silica. Upon pyrolysis of KHC4 at 600 °C, as seen in Figure 8a, the amorphous silicon dioxide transforms into cristobalite (main reflection with an interlayer distance of 4.07 Å). After vapor-gas activation of this sample, other cristobalite reflections also appear in its X-ray diffraction pattern (Figure 8b). The intensity of the SiO_2_ phase reflections is significantly reduced as a result of the reduction in silica after alkaline treatment of the samples. However, quartz reflections with an interlayer distance of 3.35–3.37 Å are recorded in the X-ray diffraction patterns of the samples after combined treatment with alkali and water vapor, regardless of the treatment sequence (Figure 8c,d). At a higher temperature treatment (>1000 °C), silicon carbide is formed, as evidenced by the 2.51–2.55 Å reflections on the X-ray diffraction patterns of samples KHC4-600A-1000 and KHC4-600A-1650 (Figure 8e,f). During the processes under consideration, carbon structure ordering is also observed. While the KHC4-600 sample has an interlayer distance (d_002_) of 3.77 Å, KHC4-600V has d_002_ = 3.65 Å, while KHC4-600VA and KHC4-600AV have d_002_ = 3.64 Å. As previously shown [54], the KHC4-600A-1000 and KHC4-600A-1650 samples have a more ordered component with d_002_ = 3.43 Å and 3.37 Å and crystallite sizes of 70–90 Å (L_c_) and 70 Å (L_a_), respectively, in addition to the weakly ordered graphite-like phase with d_002_ = 3.67 Å and 3.68 Å, respectively. Thus, the graphitization process occurs during the high-temperature treatment. However, the conditions created were obviously unsuitable for obtaining a two-dimensional crystal.

### 3.5. Structure and Morphology of Cellulose-Based Materials Particles

Kürschner and Hoffer cellulose does not change its surface topology when exposed to a temperature of 600 °C during pyrolysis. After chemical activation with the alkaline solution, the siliceous shell is destroyed [65]. Figure 9a shows a general view of the product of pyrolysis and combined activation of cellulose isolated from rice husk by the Kürschner and Hoffer method. At a higher magnification (Figure 9b), it can be seen that a porous structure is formed under the influence of water vapor and sodium hydroxide solution. The pore size predominantly varies from 1.5 to ~4.5 μm.

As determined by TEM, KHC4-600 is formed by large plate-shaped conglomerates with a pronounced layering direction (Figure 10a). The conglomerates are built up by rounded particles of about 30–50 nm in size (Figure 10b). Microdiffraction patterns taken from different particles (shown in Figure 10c, for example) differ in the intensity and position of the rings. In general, it can be concluded from their character that carbon particles have some order in volume. Based on the position of rings in the microdiffraction patterns, the following d_002_ values were determined for different carbon particles: 3.17, 3.63, 3.53 Å, etc. KHC4-600A particles also have a similar structure. However, for this sample, it was found that the lamellar formations are layer upon layer composed of small flattened particles or covered with granular particles with a diameter of 15–30 nm and finer ones of 6–8 nm and have pores of about 5–12 nm [65]. No tubular-form carbon found in rice husk carbonizates [52] or carbon fibers found in rice husk lignin carbonizates [50] were identified among any tested rice husk cellulose-based samples.

During microdiffraction studies, the presence of substances with crystal packing was also detected in the sample after alkaline treatment. Based on the set of interlayer distances in the microdiffraction patterns (0.452, 0.416, 0.274, 0.148, and 0.118 nm), we can assume the existence of various silica-containing substances, more likely in the form of silicon dioxide but possibly also as silicon carbide. The presence of a silica grain in sample KHC4-600A was confirmed by micrographs taken under an optical microscope (Figure 10d). These micrographs show that the size of the quartz grain is comparable to the size of the carbonized cellulose fragments. Carbonaceous matter was represented in several forms. Unconverted forms of carbon were also recorded.

### 3.6. Textural Characteristics of Cellulose-Based Materials

Figure 11a,b shows the nitrogen adsorption–desorption isotherms of different samples produced from rice husk cellulose. The analysis shows that they belong to type IV isotherms according to the BDDT classification [66]. The presence of a hysteresis loop indicates the existence of a continuous pore system [67]. The shape of the hysteresis loop, as noted in paper [68], indicates the formation of predominantly slit-shaped pores, although it is believed [67] that other types of pores may also be present. The curved character of the initial section of the isotherm indicates a strong adsorbate–adsorbent interaction. The pore radius (R) varies depending on the method of activation (Figure 12). In the case of carbonizate activation with water vapor (KHC4-600V), the average pore radius is 1.66 nm. When carbonizate is treated with alkali (KHC4-600A), larger pores (R = 2.5 nm) are formed, mainly due to the removal of silica as well as alkali-soluble hydrocarbon compounds formed during pyrolysis. A similar effect is observed when treating the sample initially subjected to water vapor activation with alkali. Using KHC4-600VA as an example, it can be seen that the pore radius increases from 1.66 nm to 2.05 nm. Activation by water vapor after alkaline treatment (KHC4-600AV) naturally promotes the formation of thinner pores (R = 2.34 nm vs. R = 2.5 nm in the sample KHC4-600A), and high-temperature treatment (KHC4-600A-1000 and KHC4-600A-1650) causes their enlargement (R = 2.54–2.81 nm). All types of activation promote the growth of the total pore volume (V) and specific surface area (S) of rice husk cellulose carbonizate. However, these parameters reach their maximum values in the sample KHC4-600VA (1.22 cm^3^·g^−1^ and 1197 m^2^·g^−1^, respectively).

Treatment at high temperatures (1000 and 1650 °C) is a very energy-intensive method. Furthermore, it did not result in the expected effect, i.e., the formation of carbon nanotubes or graphene-like structures was not observed. Therefore, it seems inappropriate to use this method to obtain a sorbent.

There is a widespread belief that the higher the specific surface area of the sorbent, the higher its sorption activity. In this connection, the KHC4-600VA sample with the maximum specific surface area was assumed to be a good sorbent. Moreover, it had the largest total pore volume. Therefore, this sample was chosen to be tested for the sorption of Re (VII) ions.

### 3.7. Rhenium Adsorption Process

Initially, sorption was carried out using a model solution with a rhenium concentration of 10.9 mg·dm^−3^ and a pH~6, as described in Section 2.4, and varying the sorbent-solution contact time from 30 to 200 min. The adsorbent amount and solution volume remained constant throughout the experiment. The obtained results showed (Figure 13a) that the investigated sorbent provides extraction of rare metal ions at a level of almost 90%. With increasing contact time, the degree of rhenium extraction slightly increases already after the first 30 min of the sorbent-solution interaction. Figure 13b shows a micrograph of the sorbent after the sorption process. As can be seen in the figure, the surface of the sorbent looks similar to the one presented in Figure 9a. Other researchers have also noted the analogy of plant sorbent surfaces before and after the adsorption of metals [69].

The influence of the initial concentration of rhenium ions in the range from 20 to 100 mg·dm^−3^ was investigated at pH~6 and sorbent-solution contact times of 30 and 60 min. The choice of this range of Re (VII) initial concentrations and pH is explained by the operating parameters of the technology developed to process local man-made rare metals. In addition, according to papers [27,44,46,47], the pH range of 5–6 is preferable to provide REEs with maximum adsorption by different kinds of sorbents. It is shown (Figure 14 and Figure 15) that in the case of 60 min, the process of rhenium extraction from solutions with different concentrations proceeds more efficiently. The concentration of Re (VII) ions in the mass of the KHC4-600VA sorbent increases (Figure 14), but the percentage of metal extraction slightly decreases (Figure 15) with increasing the initial solution concentration, regardless of the sorbent-solution contact time. This is obviously due to the known fact [70] that as the concentration of metal in the sorbent phase increases, the sorbent’s affinity for the metal ion decreases. However, this trend is observed to a lesser extent at a contact time of 60 min. In general, the 60-min isotherm of Re (VII) removal (Figure 14) can be classified as the C curve according to the Giles isotherm classification [71]. It means that the number of sites on the KHC4-600VA surface for Re (VII) adsorption remains constant at all concentrations up to saturation.

The most common Langmuir (1) and Freundlich (2) isotherm models were used to quantitatively describe the equilibrium of Re (VII) sorption by the KHC4-600VA sorbent for 60 min [72]:(1)$$a=\frac{a_{m}\cdot K_{L}\cdot c}{1+K_{L}\cdot c},$$
where *a* is the adsorption magnitude (mg·g^−1^); *a_m_* is the adsorption monolayer capacity or maximum adsorption (mg·g^−1^); *c* is the equilibrium concentration of the dissolved substance (mg·dm^−3^); and *K_L_* is the adsorption equilibrium constant.
(2)$$a=K_{F}\cdot c^{1/n},$$
where *a* is the adsorption magnitude (mg·g^−1^); *c* is the equilibrium concentration of the dissolved substance (mg·dm^−3^); and *K_F_* and *1*/*n* are empirical parameters, constants for the adsorbent and the adsorbate at a given temperature.

Figure 16 and Figure 17 show the Langmuir and Freundlich isotherms, and Table 2 summarizes the corresponding parameters of the studied process.

The parameters of the Freundlich isotherm, according to the correlation coefficient R^2^ (Figure 17, Table 2), correspond best to the experimental data of Re (VII) adsorption by the KHC4-600VA sorbent. It is believed [72,73] that the Langmuir equation is fulfilled for solutions with low sorbate concentrations and describes monolayer adsorption. The Freundlich isotherm equation is applicable to multilayer adsorption on a heterogeneous surface with a non-uniform distribution of adsorption heat. The value of the constant *K_F_* included in the Freundlich equation is a quantitative measure of the sorbed ion’s affinity to the adsorbent surface. In other words, *K_F_* characterizes the sorption capacity, and the parameter *1*/*n* characterizes the adsorption intensity [73,74]. The higher the value of *n* and the smaller *1*/*n*, the stronger the affinity of the sorbed ion to the adsorbent surface and the higher the heterogeneity of the adsorption process. If *n* lies between the values of 1 and 10, it means that adsorption proceeds favorably [72,73,74]. Accordingly, it can be concluded that the studied adsorption of Re (VII) proceeds favorably on the heterogeneous surface of the KHC4-600VA sorbent.

## 4. Discussion

A cellulose material was obtained by processing multi-tonnage waste from rice production using various methods. After delignification of rice husk by Beckman’s method, up to 15.5% lignin is retained in its composition while achieving maximum reduction in silica (up to 0.1%). Treatment with a nitric acid solution in ethyl alcohol according to the method of Kürschner and Hoffer provides complete delignification of the initial plant raw material. However, up to 25.2% of silicon dioxide is retained in the composition of the resulting cellulose. When comparing the IR spectra of AC, KHC4, and the initial RH [52], it is found that the character of the absorption bands is slightly different, mainly in the range of 1700–400 cm^−1^. In the IR spectrum of AC (Figure 2a), the ratio of the intensities of the absorption bands at wavenumbers 1430 cm^−1^ and 900 cm^−1^ decreases, which may indicate either the conversion of cellulose I to cellulose II or the conversion of crystalline cellulose to amorphous cellulose [62,75]. The bands typical for crystalline cellulose undergo changes [58]: the band at 1320 cm^−1^ disappears, and the band at 1375 cm^−1^ (deformation vibrations of the CH-group) shifts to the high-frequency range (1384 cm^−1^), while its intensity significantly decreases. The intensity of the band at 1460 cm^−1^ (deformation vibrations of the CH_2_-group) decreases. In the IR spectrum of KHC4, the band at 1320 cm^−1^ is retained as noted above, and the band in the range of 900 cm^−1^ disappears. However, there is a change in absorption in the 3500–2900 cm^−1^ range (valence vibrations of the CH_2_–OH groups). Since the IR transmission spectrum reflects the structural characteristics of macromolecules in the polymer volume [76], the detected changes can be considered an indicator of a decrease in the structural ordering of cellulose during its derivation from rice husk, in which it is present in a highly crystalline state [56]. Moreover, sodium hydroxide delignification results in the most significant loss of molecular structure order in isolated cellulose.

The thermal decomposition of cellulose reaches its maximum at 260 °C. Destructive-condensation processes during the carbonization of organic matter end at 625 °C. In this regard, a temperature of 600 °C was chosen as the temperature of cellulose pre-pyrolysis in order to obtain carbon sorbents. Depending on the method of activation of rice husk cellulose carbonizate by physical, chemical, and thermal methods, different functional groups are formed on the surface of the activated products, as evidenced by TPD-MS experiments. The highest amount of volatiles released during TPD-MS analysis was recorded for the KHC4-600A sample (Figure 6a). This is explained by the interaction of alkali adsorbed on the surface of carbonized cellulose with hydrocarbon compounds formed during the carbonization of Kürschner and Hoffer cellulose and condensed in the pores of the carbonizate. CO, H_2_O, CO_2_, and H_2_ were identified as volatile compounds (Appendix A). It is known that the release of CO and CO_2_ characterizes the decomposition process of oxygen-containing functional groups [77]. According to the TPD curves (Appendix A) plotted against the ion signals in the mass spectra (Appendix A), the CO_2_ release falls in the low-temperature range in all analyzed samples. This may be an indicator of carboxylic acid decomposition [77]. Conversely, the CO release is predominant in the high-temperature range, apparently as a result of the decomposition of carbonyl, phenols, ethers, and pyrone groups [77]. The hydrogen release proceeds similarly to the CO release process, and its amount also increases markedly in the high-temperature range. The highest amount of CO and H_2_ produced is from the KHC4-600A sample (Appendix A). This is followed by the KHC4-600VA and KHC4-600AV samples in terms of the amount of volatile compounds (Figure 6c,d) and the abovementioned gases (Appendix A). The carbon structure of these samples, according to X-ray phase analysis, is characterized by the same degree of ordering since they have the same values of the interlayer distance of the graphite-like structure (d_002_ = 3.64 Å). As a result of combined water vapor and chemical activation, the presence of quartz with an interlayer distance of 3.35–3.37 Å was recorded in these samples (Figure 8c,d). Quartz grains were also observed in the KHC4-600A sample using an optical microscope (Figure 10d). It is quite possible that due to the presence of some silica in the form of quartz, it cannot be completely removed during the alkaline activation of cellulose carbonizate, as evidenced by the results of XRD analysis and TEM with microdiffraction. Accordingly, during the high-temperature treatment, due to the close contact between the carbon and silicon dioxide phases, silicon carbide formation (2.51–2.55 Å reflections in the X-ray diffraction patterns, Figure 8e,f) occurs already at 1000 °C, although this process is traditionally believed to occur at higher temperatures. In paper [78], in the study of silicon conversion as a function of temperature, pressure, and the amount of carbon reducing agent, it was shown that, under certain conditions, the formation of the SiC phase is possible at relatively low temperatures (1127 °C).

When studying the porous structure of the obtained samples by TEM and BET nitrogen adsorption–desorption methods, a good agreement between the obtained data is observed. The pore size was found to be 5 nm. It was determined that water vapor activation promotes the formation of thinner pores, while alkali treatment induces the formation of larger pores. A similar trend in the pore size of sorbents was observed by Hadi et al. [72], comparing the effect of physical and chemical activation methods in obtaining active carbons. All the tested activation methods have a positive effect on the textural characteristics of the obtained sorbents. Three samples have the most developed specific surface area and the largest volume of sorbing pores: KHC4-600A, KHC4-600VA, and KHC4-600AV. This is explained by the fact that, due to the presence of silica in the composition of cellulose carbonizate, combined activation methods are the most effective. However, in terms of textural characteristics, the KHC4-600VA sample (S = 1197 m^2^·g^−1^, V = 1.22 cm^3^·g^−1^, and R = 2.05 nm) appears to be the best. Evidently, for rice husk cellulose, the combined activation technique with chemical treatment after vapor-gas activation is the most preferable for creating a developed surface among all tested methods. As for high-temperature (1000 and 1650 °C) treatment, it is not possible to recommend it for sorbent production because it is too energy-intensive.

The KHC4-600VA sorbent showed itself well enough in the process of rare metal sorption using Re (VII) ions as an example. During the first 30 min of the interaction with the solution, it extracts more than 80% of the metal ions, and at longer contact times, it extracts ~90%. This process is best described by the Freundlich isotherm equation (R^2^ = 0.9745). The sorption proceeds favorably. The energy of the KHC4-600VA sorbent interaction with rhenium ions, based on the comparison of the adsorption equilibrium constant *K_L_* (0.290) calculated by the Langmuir equation (Table 2) and the data presented in paper [47], is higher than the energy *K_L_* (0.148) of the sorbent obtained in the same way from rice husk carbonized at 400 °C. It is known that the adsorption process depends on a number of factors. In addition to textural characteristics, the evaluation of acid-base properties and surface charge is important. The type of adsorbate is of great importance [72], as is the possibility of reusing the sorbent after the desorption process. In this regard, the research work on the created sorbent will be continued to study the mentioned factors. This will allow for objective preparation for the testing of the new sorbent under pilot conditions and its subsequent recommendation for production.

## 5. Conclusions

Multi-tonnage waste from rice production was processed to obtain cellulose as a feedstock for the production of active carbon sorbents. It is shown that in the case of cellulose production by the method of Kürschner and Hoffer, the action of nitric acid in an alcoholic solution provides complete delignification of plant raw materials. At the same time, the ordering of the molecular structure of the obtained cellulose decreases to a lesser extent as compared to delignification with sodium hydroxide solution according to Beckman’s method. The silicon dioxide contained in the rice husk is transferred into Kürschner and Hoffer cellulose composition.

During the pyrolysis and subsequent activation of Kürschner and Hoffer cellulose by different methods, structural transformations of its organic and mineral components take place. During pyrolysis of KHC4 at 600 °C, a graphite-like structure with an interlayer distance of 3.77 Å is formed. Amorphous silicon dioxide transforms into cristobalite. There are quartz grains, so it is not possible to completely remove silicon dioxide during the activation of cellulose carbonizates with alkali. Water vapor activation, carried out independently and combined with chemical activation, promotes the reduction in the interlayer distance in carbon crystallites of activated materials to 3.64–3.65 Å. The silicon carbide phase is formed during high-temperature (at 1000 °C and above) treatment.

Taking into account the carbon–mineral composition of cellulose carbonizate isolated from rice husk by the Kürschner and Hoffer method, the most effective are combined activation methods. The technical way of chemical treatment with alkali after water vapor activation allows for the obtaining of the KHC4-600VA sorbent with the following characteristics: specific surface area ~1200 m^2^·g^−1^, total pore volume 1.22 cm^3^·g^−1^, and pore radius 2.05 nm. The KHC4-600VA sorbent provides 90% extraction of Re (VII) ions from aqueous solutions. Sorption proceeds favorably and is described by the Freundlich isotherm equation.

## Figures and Tables

**Figure 1 materials-16-06277-f001:**
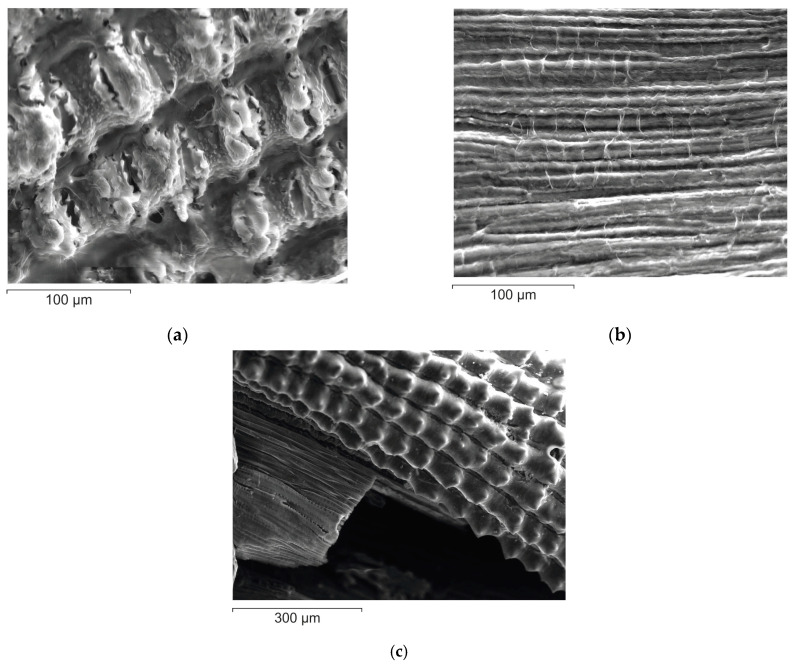
SEM images of rice husk cellulose: (**a**) outer side of AC; (**b**) inner side of AC; (**c**) outer and inner sides of KHC4.

**Figure 2 materials-16-06277-f002:**
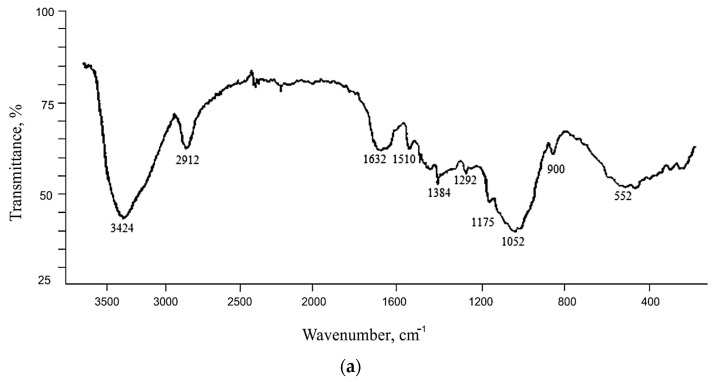
IR spectra of rice husk cellulose: (**a**) AC; (**b**) KHC4.

**Figure 3 materials-16-06277-f003:**
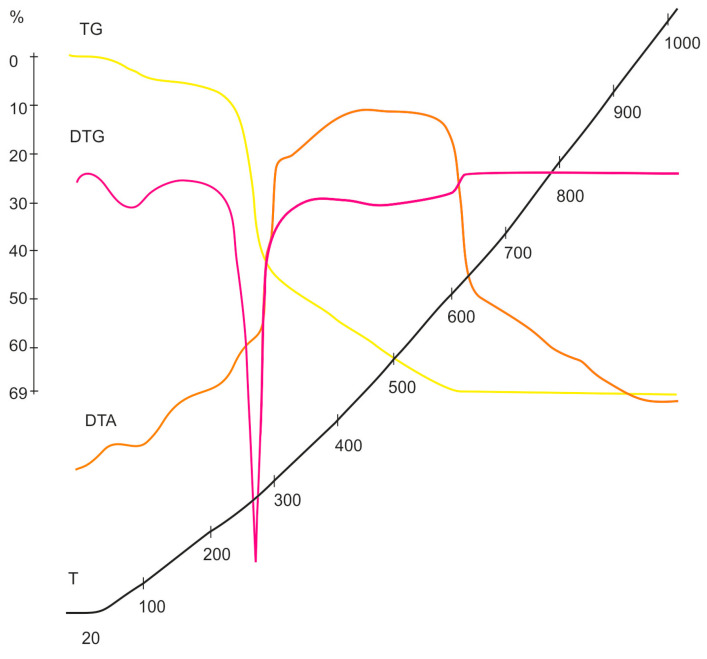
KHC4 derivatograms.

**Figure 4 materials-16-06277-f004:**
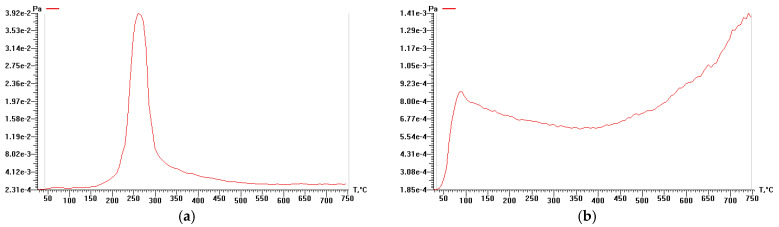
P-T curves of rice husk cellulose and its carbonizate (on Y-axis—“me−n” means “m × 10^−n^”): (**a**) KHC4; (**b**) KHC4-600.

**Figure 5 materials-16-06277-f005:**
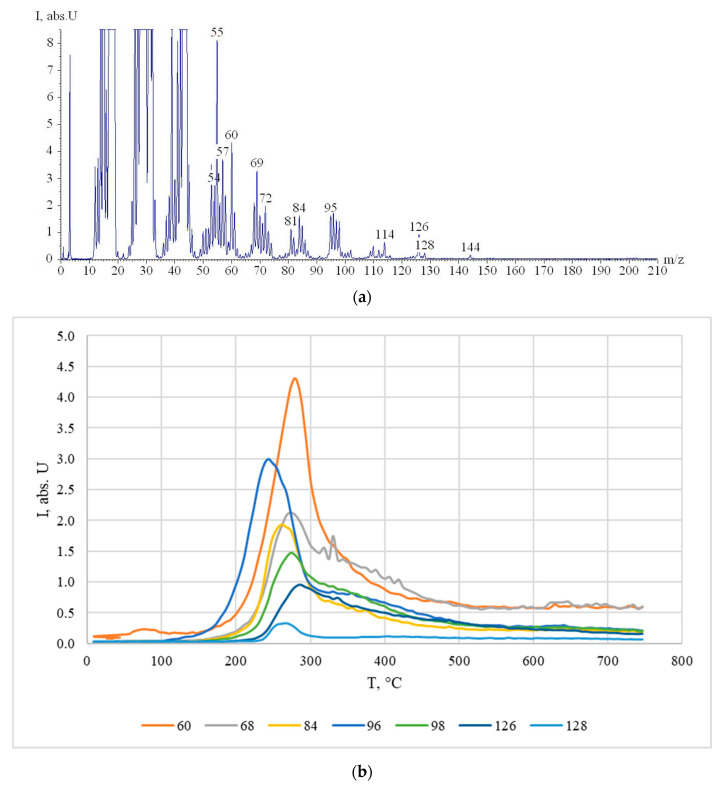
TPD-MS results: (**a**) mass spectrum of KHC4 pyrolysis products at 280 °C obtained after electron ionization; (**b**) TPD-curves of ions with *m*/*z* 60, 68, 84, 96, 98, 126, and 128 under pyrolysis of KHC4.

**Figure 6 materials-16-06277-f006:**
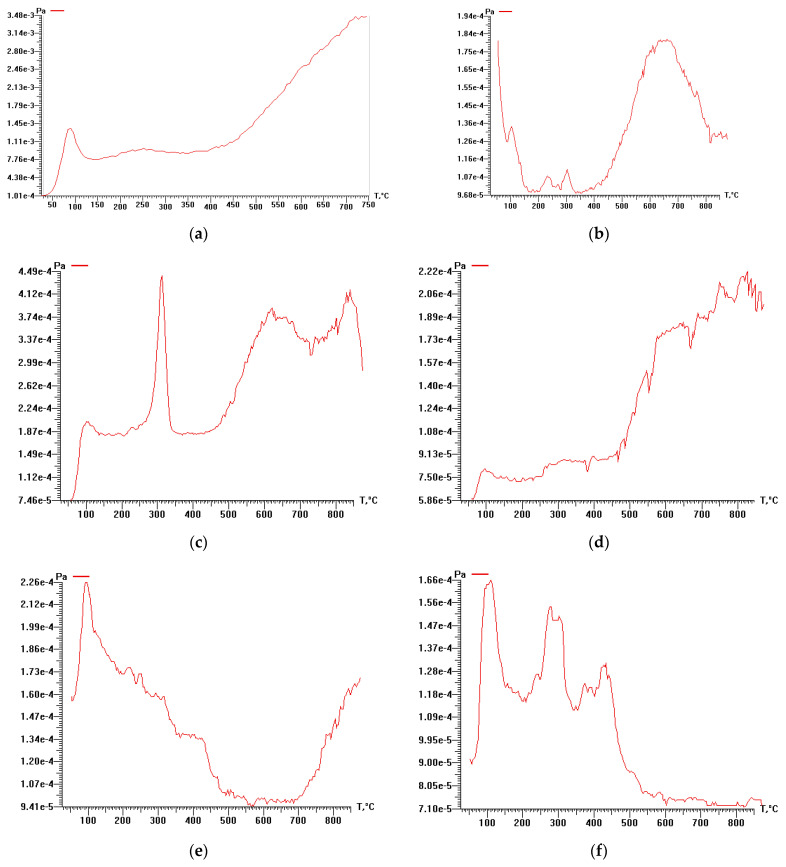
P-T curves of cellulose-based materials (on Y-axis—“me−n” means “m × 10^−n^”): (**a**) KHC4-600A; (**b**) KHC4-600V; (**c**) KHC4-600VA; (**d**) KHC4-600AV; (**e**) KHC4-600A-1000; (**f**) KHC4-600A-1650.

**Figure 7 materials-16-06277-f007:**
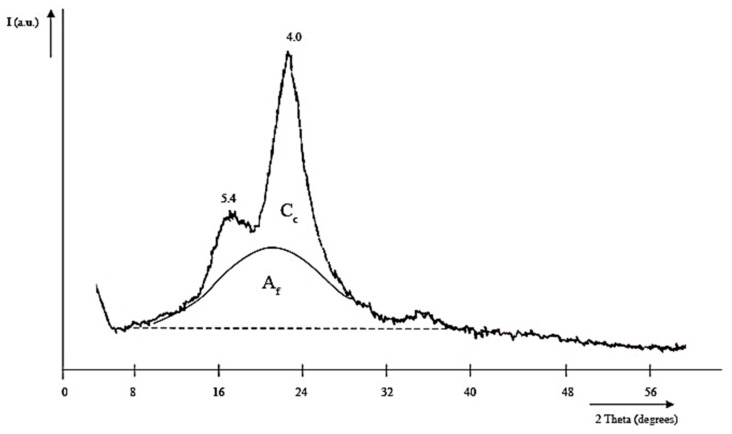
KHC4 sample X-ray diffraction pattern.

**Figure 8 materials-16-06277-f008:**
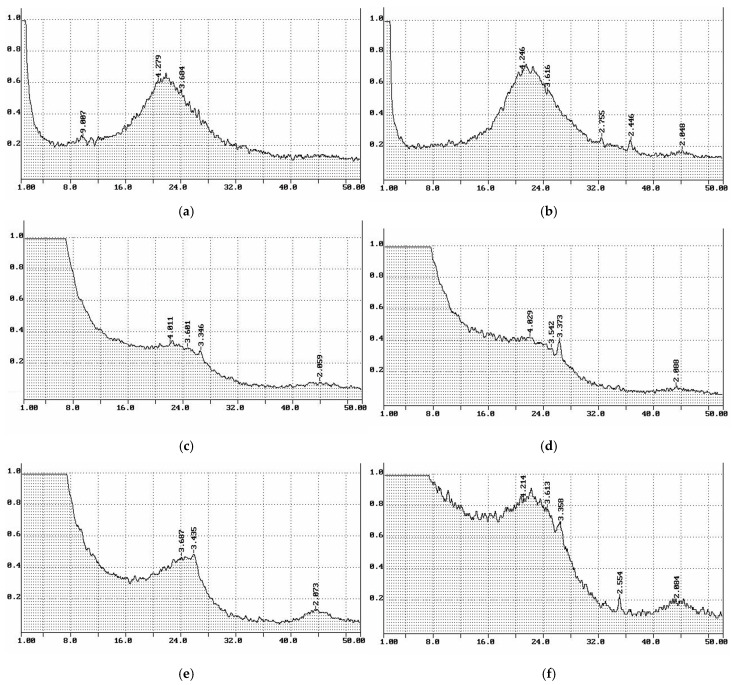
Cellulose-based materials X-ray diffraction patterns (on X-axis—2Θ, degrees; on Y-axis—Intensity, Abs.U.): (**a**) KHC4-600; (**b**) KHC4-600V; (**c**) KHC4-600VA; (**d**) KHC4-600AV; (**e**) KHC4-600A-1000; (**f**) KHC4-600A-1650.

**Figure 9 materials-16-06277-f009:**
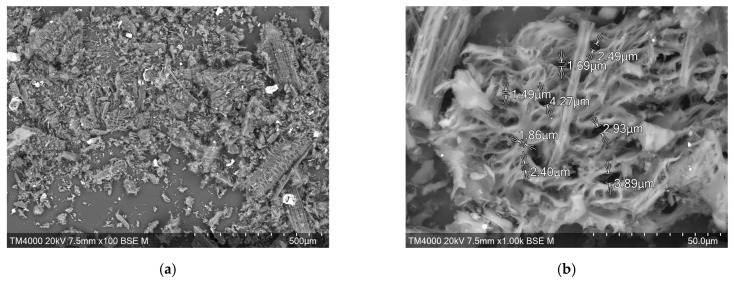
SEM images of the KHC4-600VA sample: (**a**) total view; (**b**) pore structure.

**Figure 10 materials-16-06277-f010:**
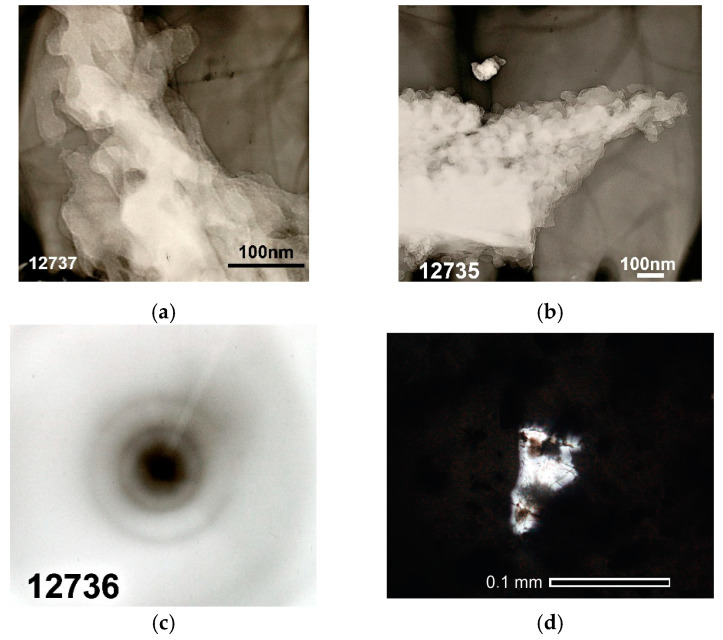
Micrographs and microdiffraction patterns of the cellulose-based materials: (**a**) TEM micrograph of the KHC4-600 sample of a large conglomerate; (**b**) TEM micrograph of the KHC4-600 sample of the structure of a large conglomerate; (**c**) microdiffraction pattern from the particles shown in (**b**); (**d**) optical micrograph of the KHC4-600A sample in direct transmitted light, Nicoli II.

**Figure 11 materials-16-06277-f011:**
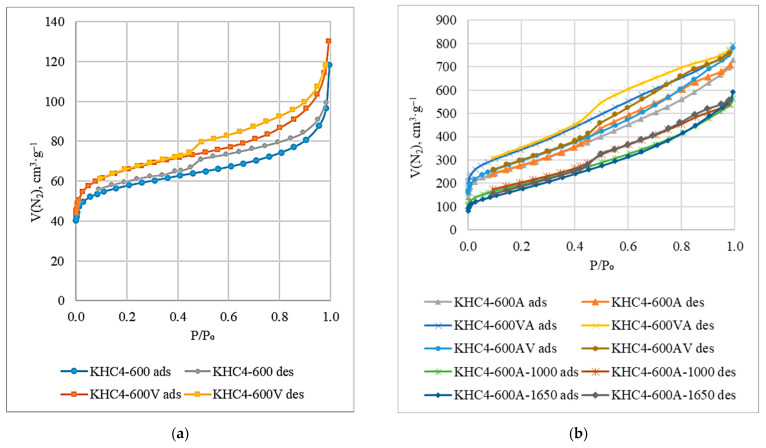
N_2_ adsorption–desorption isotherms of the cellulose-based sorbents: (**a**) KHC4-600 and KHC4-600V; (**b**) KHC4-600A, KHC4-600VA, KHC4-600AV, KHC4-600A-1000, and KHC4-600A-1650.

**Figure 12 materials-16-06277-f012:**
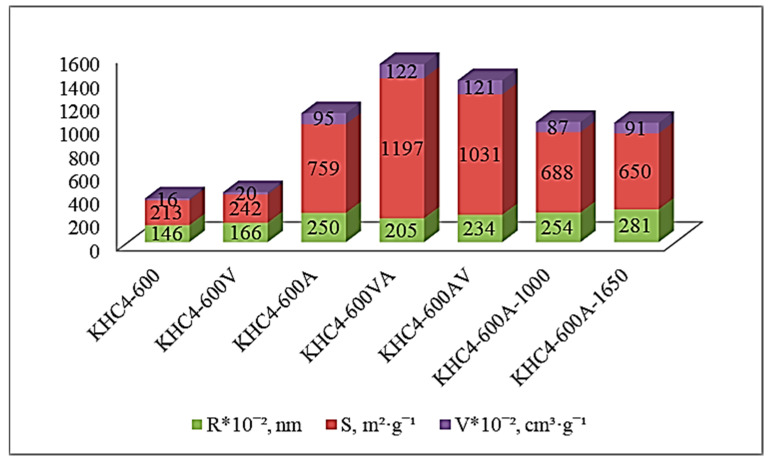
Textural characteristics of cellulose-based materials.

**Figure 13 materials-16-06277-f013:**
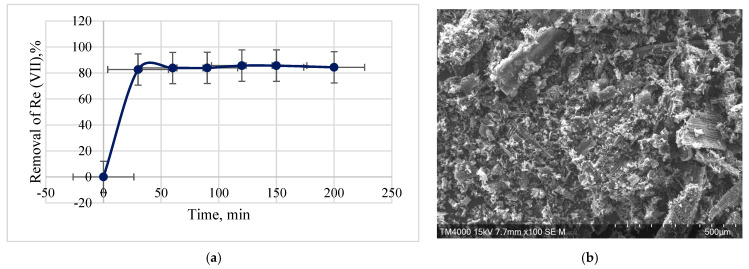
Removal of Re (VII) from the solution with its initial concentration of 10.9 mg·dm^−3^ by the KHC4-600VA sorbent: (**a**) Re (VII) ions removal for different contact times; (**b**) SEM image of the KHC4-600VA sorbent after the sorption process.

**Figure 14 materials-16-06277-f014:**
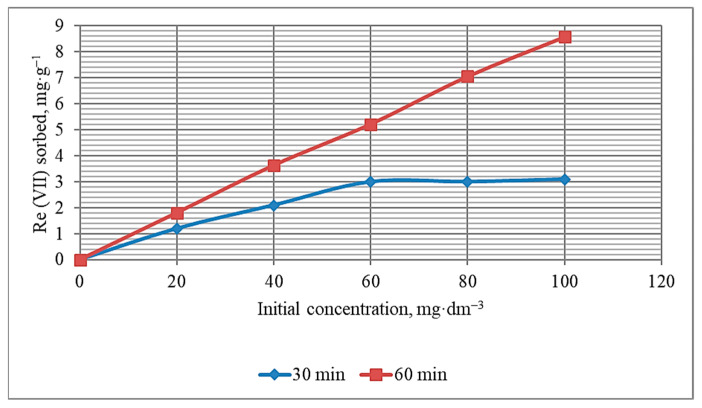
Sorption of Re (VII) from solutions with different initial concentrations by the KHC4-600VA for 30 and 60 min.

**Figure 15 materials-16-06277-f015:**
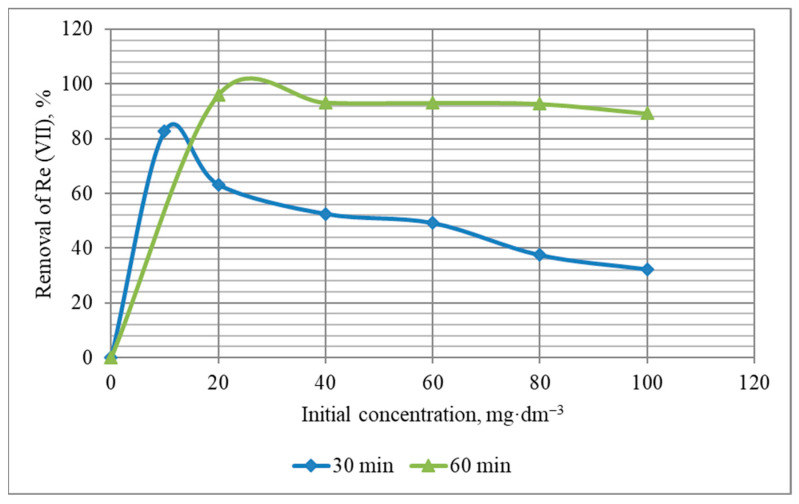
Removal of Re (VII) from solutions with its different initial concentrations by the KHC4-600VA for 30 and 60 min.

**Figure 16 materials-16-06277-f016:**
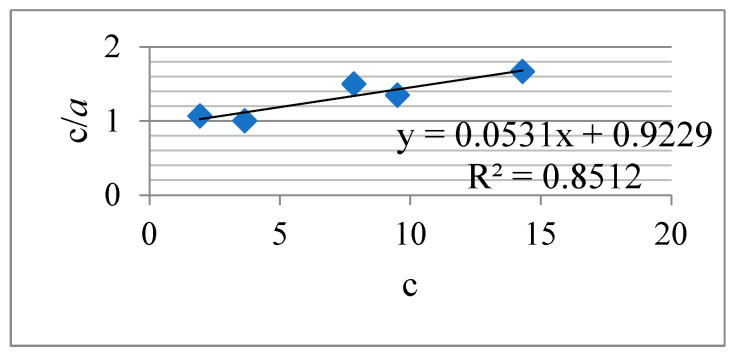
Langmuir isotherm for the Re (VII) sorption by the KHC4-600VA in 60 min.

**Figure 17 materials-16-06277-f017:**
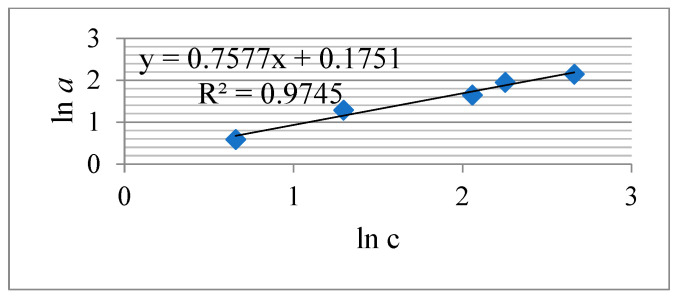
Freundlich isotherm for the Re (VII) sorption by the KHC4-600VA in 60 min.

**Table 1 materials-16-06277-t001:** Yield and composition of rice husk delignification products.

Sample	Delignification Conditions	Yield, % wt.	Content, % wt.			
			Cellulose	Hemicellulose	Lignin	SiO_2_
AC	According to Beckman’s method	43.4	62.0	22.4	15.5	0.1
KHC2	2-fold treatment with Kürschner and Hoffer reagent	53.1	not determined	not determined	~1.0	26.4
KHC3	3-fold treatment with Kürschner and Hoffer reagent	50.0	not determined	not determined	N/A	27.4
KHC4	4-fold treatment with Kürschner and Hoffer reagent	49.6	not determined	not determined	N/A	25.2

**Table 2 materials-16-06277-t002:** Langmuir and Freundlich parameters for the Re (VII) sorption by the KHC4-600VA in 60 min.

Langmuir constants	R^2^	*a*_m_, mg·g^−1^	*K_L_*, dm^3^·mg^−1^
	0.8512	4.01	0.290
Freundlich constants	R^2^	*K_F_*, (mg·g^−1^)·(dm^3^·mg^−1^)^1/n^	1/*n*
	0.9745	1.221	0.510

## Data Availability

Not applicable.

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
