# Peer review of "Rice Husk Cellulose-Based Adsorbent to Extract Rare Metals: Preparing and Properties"

_materials, 2023, doi:10.3390/ma16186277_

Round 1

Reviewer 1 Report

The paper on preparing and characterizing rice husk cellulose-based adsorbent is a complete and well-written document. However, the rhenium adsorption part is limited. 

Comments and Suggestions for Authors:

1. In Figure 2a, I suggest improving the presentation because the wavelengths are not adequately visualized; they overlap with the spectrum.

2. Improve the presentation of Figure 3 (KHC4 derivatograms) and use colors to facilitate the analysis.

3. In Figure 5, remove the decimals from the axes to make them uniform with those of the other figures.

4. In Figure 11 (N2 adsorption-desorption isotherms), limit the X-axis to relative pressures 1.0. Improve the presentation of Figure 11b since the isotherms of the samples are not differentiated.

5. The information in Figure 12 (Textural characteristics of cellulose-based materials) could be put into a table to facilitate the analysis. The information in the Figure needs to be clarified.

6. Figure 13 should include error bars.

7. In Figure 14 (Sorption of Re (VII) from solutions with its different initial concentrations by the KHC4600VA for 30 and 60 min), it is necessary to include more experimental data for analysis. I suggest comparing the isotherms obtained with contact times of 60 min with the Giles isotherm classification.

8. The justification for selecting the KHC4-600VA material as Re(VII) adsorbent should be included since it is only mentioned that it presented the highest specific surface area. What effect does the pH of the solution have on the adsorption process? What is the pHzpc of the adsorbent?

9. The fit of the adsorption isotherm data to the Langmuir and Freundlich models should be done with nonlinear methods, and the parameters include the deviations. 

10. It would be interesting to include desorption and adsorbent reuse analyses.

Author Response

Dear Reviewer,

Thank you very much for your time and efforts to review the manuscript. Your comments have been very useful, helping us to improve it. Please find the detailed responses attached. Pictures have been improved and changed in the paper. The text information added in the manuscript according to your comments was highlighted in red and violet.

Many thanks!

Best regards,

Prof. Svetlana Yefremova

Author Response

Dear reviewer,

Authors would like to thank you for your reviewing and high estimate of our manuscript.

It took a lot of your time to discuss the paper in detail. We tried to pay attention to each your comments and recommendations which were useful to improve it.

Please see our response attached. Changes in the article according to your comments was highlighted in blue and violet.   

Thank you again.

Best regards,

Prof. Svetlana Yefremova
